# Inhibitor of Growth Factors Regulate Cellular Senescence

**DOI:** 10.3390/cancers14133107

**Published:** 2022-06-24

**Authors:** Soudeh Ghafouri-Fard, Mohammad Taheri, Aria Baniahmad

**Affiliations:** 1Department of Medical Genetics, School of Medicine, Shahid Beheshti University of Medical Sciences, Tehran 19835-35511, Iran; s.ghafourifard@sbmu.ac.ir; 2Institute of Human Genetics, Jena University Hospital, 07743 Jena, Germany; mohammad.taheri@med.uni-jena.de

**Keywords:** ING1, ING2, ING3, ING4, ING5, cellular senescence, cancer, tumor suppressor, oncoprotein, inhibitor of growth

## Abstract

**Simple Summary:**

Five members of the Inhibitor of Growth (ING) family share a highly conserved plant homeodomian with affinity to the specific histone modification H3K4me3. Since some ING family members are preferentially associated with histone acetyltransferaseactivity while other members with histone deacetlyse activity, the ING family membres are epigenetic regulators. Interestingly, ING members can regulate the induction cellular senescence in both primray untransformed human cells as well as human cancer cells. We discuss here the up-to-date knowledge about their regulatory activity within the cellular senescent program.

**Abstract:**

The Inhibitor of Growth (ING) proteins are a group of tumor suppressors with five conserved genes. A common motif of ING factors is the conserved plant homeodomain (PHD), with which they bind to chromatin as readers of the histone mark trimethylated histone H3 (H3K4me3). These genes often produce several protein products through alternative splicing events. Interestingly, ING1 and ING2 participate in the establishment of the repressive mSIN3a-HDAC complexes, whereas ING3, ING4, and ING5 are associated with the activating HAT protein complexes. In addition to the modulation of chromatin’s structure, they regulate cell cycle transition, cellular senescence, repair of DNA damage, apoptosis, and angiogenic pathways. They also have fundamental effects on regulating cellular senescence in cancer cells. In the current review, we explain their role in cellular senescence based on the evidence obtained from cell line and animal studies, particularly in the context of cancer.

## 1. Introduction

The Inhibitor of Growth (ING) proteins constitute a family of tumor suppressors with five conserved genes in humans and mice, most of them producing several protein products through alternative splicing events [1,2]. Figure 1 shows the genomic organization and alternatively spliced variants of the human ING family members.

The first two members of this family have been shown to participate in the establishment of mSIN3a-HDAC complexes. In addition, ING1, ING4, and ING5 have functional associations with HAT protein complexes, indicating that ING1 has an opposing epigenetic activity. These proteins regulate cell cycle transition, cellular senescence, repair of DNA damage, apoptosis, angiogenic pathways, and chromatin’s structure [1]. ING3 associates with the TIP60 complex, a histone acetyltransferase. Although ING1 and ING2 can associate with HAT activity [3], they are mainly found in mSIN3A/HDAC complexes [4].

In fact, ING proteins participate in cell cycle checkpoints [5,6]. Most studies have assessed the function of ING1. However, other studies on ING2-5 have shown similar functions for these ING proteins. Furthermore, the inhibition of INGs’ expressions has been found to enhance cell migration and release cells from contact inhibition [7,8,9,10]. Moreover, the majority of ING proteins have been found to play essential roles in appropriate p53 function [5,6]. Meanwhile, p53-independent functions have been identified for INGs [7]. These proteins also participate in the establishment of chromatin-remodeling complexes [4]. Thus, they might regulate the transcription of genes in the nucleus [11,12,13]. Table 1 provides information about different ING family members, their specific and redundant functions, and related molecular pathways.

From an evolutionary point of view, orthologs of ING proteins have been detected in almost all species from yeast to humans [14]. The plant homeodomain (PHD) is a shared and conserved fundamental part of ING factors that has the ability to bind to the trimethylated lysine of histone H3 (H3K4me3). Mutations in this domain lead to defects in the induction of cellular senescence, indicating the functional link between the recognition of chromatin marks and cellular senescence [1]. As type-II tumor suppressors, these proteins have inhibitory roles in carcinogenesis. The down-regulation of the expression of ING1 or loss of heterozygosity in the corresponding locus has been detected in several types of cancers [15,16]. Moreover, the down-regulation of ING2 has been reported in a variety of tumors [16].

Evidence for the involvement of INGs in cellular senescence first came from investigations in the late 1990s. First, the expression of a certain splice variant of ING1, namely p33ING1, has been found to be elevated in senescent cells compared with young diploid fibroblasts having proliferation ability [17]. Moreover, the suppression of ING1 expression by antisense RNA has extended the replicative capacity of normal human fibroblasts, indicating the importance of p33ING1 in the induction of replicative senescence [17]. A subsequent study revealed the induction of growth arrest in primary human fibroblasts and the up-regulation of the levels of senescence-specific markers following the forced over-expression of p33ING1 in these cells [18]. p47INGa is another splice variant of ING1 whose levels have been reported to be up-regulated in the course of replicative senescence in human fibroblasts [19]. In the current review, we explain their role in cellular senescence based on the evidence obtained from cell line and animal studies, particularly in the context of cancer.

## 2. Cell Line Studies and Experiments in Clinical Samples

### 2.1. ING1

Rajarajacholan and Riabowol developed a novel cell model of ING1a-induced senescence. They reported that the ING1a epigenetic regulator can simultaneously induce senescence in mass cultures with a significantly higher speed than other modalities. It rapidly activates Rb/p16^INK4a^ to induce senescence, but it has no effect on the p53 axis. ING1a has also been shown to induce the expression of a scaffold protein with crucial effects in endocytosis, namely Intersecting 2. This leads to alterations in the stoichiometric characteristics of endocytosis proteins, resulting in the blockage of growth factor uptake and the induction of Rb signals, which results in the suppression of cell growth. Cumulatively, ING1a functions as an activator of the retinoblastoma protein (pRb) pathway that induces senescence without the induction of p53-mediated DNA damage signals [20]. Another study has indicated the up-regulation of ING1a in fibroblasts when they approach senescence. The over-expression of ING1a quickly prompts a senescent phenotype in primary diploid fibroblasts that is similar to replicative senescence by the majority of physical and biochemical procedures. This role of ING1a is mediated via the suppression of endocytosis to inhibit mitogen signaling [21]. ING1 has also been shown to interact with PCNA. This interaction is induced by ultraviolet (UV) irradiation [22]. Moreover, ING1 has functional interactions with members of the 14-3-3 family, resulting in the localization of ING1 in the cytoplasm [23]. The up-regulation of ING1 promotes Bax expression and alters the mitochondrial membrane potential through a mechanism that depends on the presence of p53 [24]. Experiments in primary fibroblasts and epithelium-derived cell lines have shown that apoptosis-inducing stimuli can increase the translocation of ING1 to the mitochondria in an independent manner from the p53 status. 

In addition to the role of ING1 in regulating cell fate in non-cancerous cells, ING1 regulates a set of different pathways in cancer cells, including the induction of cellular senescence, which inhibits the cell cycle. Notably, the apoptosis-inducing capacity of ING1 in breast cancer cell lines has been correlated with the amount of ING1 translocation to the mitochondria following exposure to UV. ING1 protein has also been shown to interact with BAX and colocalize with this protein. ING1 protein can also interact with 64 mitochondrial proteins [25]. Another study in prostate cancer (PCa) cells has shown that ING1b silencing suppresses the androgen receptor (AR)-mediated transactivation of AR targets, resulting in a reduction in the growth of these cells (Figure 2). ING1b silencing has also resulted in the enhancement of cellular senescence and induction of expression of the potent cell cycle inhibitor p16^INK4a^ in PCa cells. This unanticipated result might be due to a compensatory mechanism via the up-regulation of the ING2 levels under ING1-deficient conditions. ING2 interacts also with AR and blocks the AR-mediated activation of transcription [26]. 

Another study has indicated the up-regulation of the ING1 levels in non-cancerous senescent primary human prostate cells, suggesting that ING1 regulates cellular senescence in both non-transformed and cancerous cells. Further assays have confirmed interactions between ING1b and AR. Mechanistically, ING1b suppresses the activity of AR-responsive promoters and the expression of AR target genes in PCa cells. ING1b silencing in mouse embryonic fibroblasts has resulted in the enhancement of AR activity, signifying that interaction with ING1b suppresses AR-mediated gene transcription. Moreover, the expression of ING1b has been shown to be lower in castration-resistant prostate cancer (CRPCa) cells compared with androgen-dependent LNCaP cells. The forced up-regulation of this ING member has induced cellular senescence and reduced the migratory potential of both types of PCa cells. ING1b has also been shown to up-regulate the expression of the cell cycle inhibitor p27^KIP1^. This study demonstrated the corepressor function of ING1b in several AR-mediated activities [27].

Further evidence for the participation of ING1a in cellular senescence originated from the observed up-regulation of an alternatively spliced form of this ING member in the course of replicative senescence. The up-regulation of this ING member has inhibited cell growth, induced alterations in cell morphology, and enhanced the levels of senescence-associated β-galactosidase. Moreover, the levels of pRb, p16^INK4a^, and cyclin D1 have been increased following the up-regulation of ING1a. ING1a could also induce the expression of several genes encoding endocytosis-related proteins, particularly Intersectin 2. The up-regulation of Intersectin 2 could induce the expressions of p16^INK4a^ and p57^KIP2^, which could inhibit the inactivation of pRb. These two proteins act as downstream effectors of ING1a. The expression of Intersectin 2 is also enhanced in normally senescing cells. Senescence could also be induced by the suppression of endocytosis or alterations in the stoichiometric features of endosome constituents, including Rab family members. Moreover, Intersectin 2 has been shown to be a key transducer of ING1a-associated senescence [28]. Moreover, a series of functional assays identified a negative androgen response element in the core promoter region of hTERT. Interestingly, AR, ING1, and ING2 are recruited to that chromatin site in an androgen-dependent manner. Both ING1 and ING2 could facilitate AR-regulated transrepression. Data further suggest that AR has an oppositional, biphasic function in the regulation of the expression of hTERT, and the inhibitory effects of androgens on hTERT are dependent on the AR co-repressors ING1 and ING2 [29]. Thus, both ING1 and ING2 mediate gene repression by androgens on hTERT [29]. TERT is indirectly related to senescence as a bypass factor.

### 2.2. ING2

ING2 is another member of the ING family that can be translocated into mitochondria participating in the homeostasis of cellular metabolic pathways [30]. ING2 has also been shown to regulate the high glucose-induced cell cycle arrest and epithelial-to-mesenchymal transition (EMT) in proximal tubule epithelial cells [31]. The expression of this ING protein has been found to be reduced in osteosarcoma. Notably, a decrease in the nuclear levels of ING2 has been associated with a poor clinical outcome. Further experiments have shown the enhancement of apoptosis, cell cycle arrest, and senescence when the levels of intact ING2 have increased in cells (Figure 3) [32]. Interestingly, ING1 and ING2 crosstalk indicates a cellular back-up mechanism if one of the two INGs is malfunctioning [26]. ING2 has also been shown to be part of the mSIN3a-HDAC complex. Furthermore, it has been shown that ING2 recruits histone methyltransferase (HMT) activity with methylation site-specificity distinct from those of histone H3 lysines 4 and 9 [33]. DNA damage can initiate the recognition of H3K4me3 by the ING2 PHD domain, leading to the stabilization of the mSin3a-HDAC1 complex at the promoter regions of proliferation genes [34]. Mechanistically, it has been revealed that the epigenetic mark H3K4me3 provides the binding element for PHD that is highly conserved among the ING members. The Ala 1, Arg 2, Thr 3, and Thr 6 of the peptide have been shown to be responsible for the specificity and affinity of PHD [35]. Together with the finding that the PHD domain can bind to H3K4me3, this suggests that ING2 is recruited to active chromatin sites at H3K4me3, co-recruiting the repressive mSIN3a/HDAC complex and a currently unknown HMT to exert its effects at the epigenetic level. Moreover, ING2 can enhance the expression of PAI-1 and HSPA1A with mSin3A/HDAC1via its PHD domain and C-terminal region [36]. Finally, ING2 has been found to play crucial roles in the development of embryos via the modulation of the chromatin configuration. Its silencing in embryonic cells has resulted in the up-regulation of p21 and down-regulation of HDAC1 [37].

### 2.3. ING3

The nuclear expression of ING3 has been shown to be lower in clinical samples of head and neck squamous cell carcinoma compared with dysplasia and normal epithelium. Moreover, its expression has been found to be negatively correlated with poor differentiation and advanced TNM stages. Conversely, the cytoplasmic levels of ING3 have been elevated in tumor samples in association with lymph node metastasis and the expression of 14-3-3η. The nuclear level of ING3 has also been correlated with the expression levels of p300, p21, and acetylated p53. Thus, a reduction in the nuclear ING3 levels might be involved in the tumorigenic process. Moreover, the enhancement of cytoplasmic ING3 might result from 14-3-3η binding. This study also indicated the importance of nuclear ING3 in the enhancement of apoptosis via interaction with p300 and p21. Additionally, the interaction between ING3 and p300 can lead to the up-regulation of acetylated p53 and the induction of p53-associated cell cycle arrest and senescence [38].

ING3 was reported to act as a tumor suppressor in many different cancer types to regulate apoptosis. On the other hand, the ING3 levels are positively correlated with poor survival prognosis of PCa patients. In line with this, in PCa ING3 was shown to also have an oncogenic role by acting as a coactivator of AR [39]. Moreover, experiments in a human ex vivo prostate tissue model system identified oncogenic properties for ING3 [40]. One possibility for the differential reports could be that splice variants exhibit distinct functions. ING3 knockdown induces cellular senescence via a pathway leading to cell cycle arrest, indicating an oncogenic role for ING3 in PCa; this may be due to the ING3Δex11 splice variant lacking functional PHD in order to mediate the oncogenic characteristics by triggering EMT in PCa cells [41].

### 2.4. ING4

ING4 negatively regulates cell proliferation in normal, non-transformed primary fibroblasts. The antiproliferative action of ING4 requires its ability to recognize chromatin marks, it is p53-dependent, at least in part, and it is lost in an ING4 cancer-associated mutant [42]. However, ING4 controls the secretion of chemokines, resulting in the promotion of cancer cell proliferation. This highlights a possible dual role of ING family members analyzing intracellular or interactions with other neighboring cells. Even in nucleoli, ING factors can regulate gene expression. ING4 has been shown as a positive regulator of rRNA synthesis at the epigenetic level, which leads to the enhanced cell proliferation of the haploid HAP1 cells [43]. Moreover, ING4 has been found to modulate the proliferation of primary non-transformed human fibroblasts and coordinate a secretory feature in these cells that enhances the proliferation of malignant cells both in vitro and in animal models [44].

### 2.5. ING5

Investigations in breast cancer cells have revealed that the up-regulation of ING5 leads to a decrease in cell proliferation, reduction in glucose metabolism, induction of cell cycle arrest, reduction in the migratory potential and invasive properties, induction of apoptosis, autophagy, and senescence, and the mesenchymal–epithelial transition (MET). The association between ING5 and chemoresistance is mediated by the activation of AKT and NF-κB, up-regulation of MRP1 and GST-π, and down-regulation of FBXW7 [44]. Another study assessed the role of ING5 in cellular senescence in glioma cells. The up-regulation of ING5 in U87 cells has resulted in the suppression of proliferation, energy metabolism, migratory ability, and invasiveness; induction of cell cycle arrest at G2/M; and the enhancement of apoptosis, differentiation, senescence, MET, and resistance to a number of anticancer agents, namely cisplatin, MG132, paclitaxel, and SAHA. On the other hand, the hypoexpression of ING5 has been associated with tumorigenic processes [45]. Another study in ovarian cancer cells has shown that the up-regulation of ING5 suppresses the viability of malignant cells and reduces their glucose metabolism, migratory potential, invasiveness, and EMT. Moreover, this ING protein could induce cell cycle arrest, apoptosis, senescence, and autophagy. The impact of ING5 on chemoresistance has been associated with resistance to apoptosis and the expression of chemoresistance-related genes [46]. Finally, the truncation of ING5 in squamous cell carcinoma cells has led to the induction of senescence, but not apoptosis [47]. Table 2 shows the impact of INGs on cellular senescence based on cell line studies and investigations in clinical samples.

### 2.6. Animal Studies

Animal studies have shown that Ing1 has two alternatively spliced isoforms with opposite effects on p53 functioning. While the longer isoform (p37Ing1b) inhibits p53 functioning, the shorter ones (p31Ing1a or p31Ing1c) possibly enhance the activity of p53 [48]. A gene-targeting experiment in mouse embryonic stem cells has shown that Ing1-null mice are viable, but are susceptible to the effect of whole-body irradiation, with a proportion of them developing spontaneous cancer as they age [49]. An experiment in Ing1 knockout animal models showed that ING1 deficiency leads to the down-regulation of prostate-specific target genes of AR. Moreover, the knock-down of ING1b has resulted in the enhancement of cellular senescence [26]. ING2 deficiency has led to the activation of apoptosis mechanisms in the testis as well as the development of sarcoma in animal models [50]. Homozygous *ING3* knockout has resulted in severe growth retardation, leading to early embryonic lethality [51]. ING4 knock-out animals have exhibited a high level of morbidity and defects in the innate immune responses and angiogenesis through the hyper-activation of the NF-κB pathway [52]. Although no knockout animal has been established for ING5, it is expected that ING5 knockout would result in defects in various stem cell populations [53].

Other studies in animal models of breast cancer [44], glioma [45], and ovarian cancer [46] have shown that the over-expression of ING5 significantly reduces tumor weight and enhances apoptosis and autophagy. The impact of the over-expression or down-regulation of other members of this family on apoptosis or senescence has not been investigated in animal models (Table 3). 

## 3. Discussion

INGs are a group of tumor suppressors whose roles in cellular senescence are being elucidated. This effect is mediated through different routes, particularly epigenetic changes that modify chromatin’s structure by binding to the histone mark H3K4me3 and to histone-modifying enzyme HDAC and/or HAT protein complexes, and HMT activity. Although the effects of INGs on the induction of cellular senescence are well-established, their importance in the maintenance of this phenotype has not been clarified.

ING1 is the most-assessed member of this family in this regard. These findings were obtained from studies on cell lines and animal studies, particularly in the context of cancer. Further studies in animal models of aging-related disorders are needed for the identification of the exact functions of INGs. The results of these studies have clinical implications for the treatment of cancer and aging-related disorders. However, this field lacks sufficient evidence from human studies. Few studies in clinical samples have demonstrated abnormal expression of INGs. For instance, ING5 expression has been found to be elevated in breast cancer samples compared with normal tissues. Levels of this ING have been positively correlated with the relapse- and metastasis-free survival rates. Most notably, nuclear ING5 expression has been shown to be gradually decreased from normal breast tissue, fibroadenoma, adenomatosis, and primary cancers to metastatic cancers. On the other hand, cytoplasmic ING5 has been found to have the opposite trend. While nuclear ING5 has been negatively correlated with distant metastasis and p53 down-regulation, cytoplasmic ING5 has been positively correlated with the size of tumors and ER status [44]. These observations extend the complexity of the function and subcellular localization of INGs beyond the simple description of a tumor suppressor role. Moreover, the interactions between different members of this protein family have importance in the determination of their effects on cellular senescence.

Notably, INGs have been shown to affect the response of cancer cells to a variety of anticancer agents through the modulation of senescence or other cellular mechanisms. Therefore, these proteins represent potential targets for combating chemoresistance.

Taken together, INGs have been regarded as functional links between cancer, senescence, and apoptosis [54]. Moreover, the subcellular localization of these proteins determines their exact function. Several proteins, such as PCNA and p53, interact with INGs to affect their relocalization. Future studies are needed to determine the clinical implication of ING-targeted therapies in human disorders.

## 4. Conclusions

The ING family emerged as an interesting tool to analyze epigenetic regulation of cellular senescence in both normal immortalized cells as well as in tumors. Since cellular senescence is linked to aging and to some aging diseases future analyses of members of the ING family will shed light into novel pathways.

## Figures and Tables

**Figure 1 cancers-14-03107-f001:**
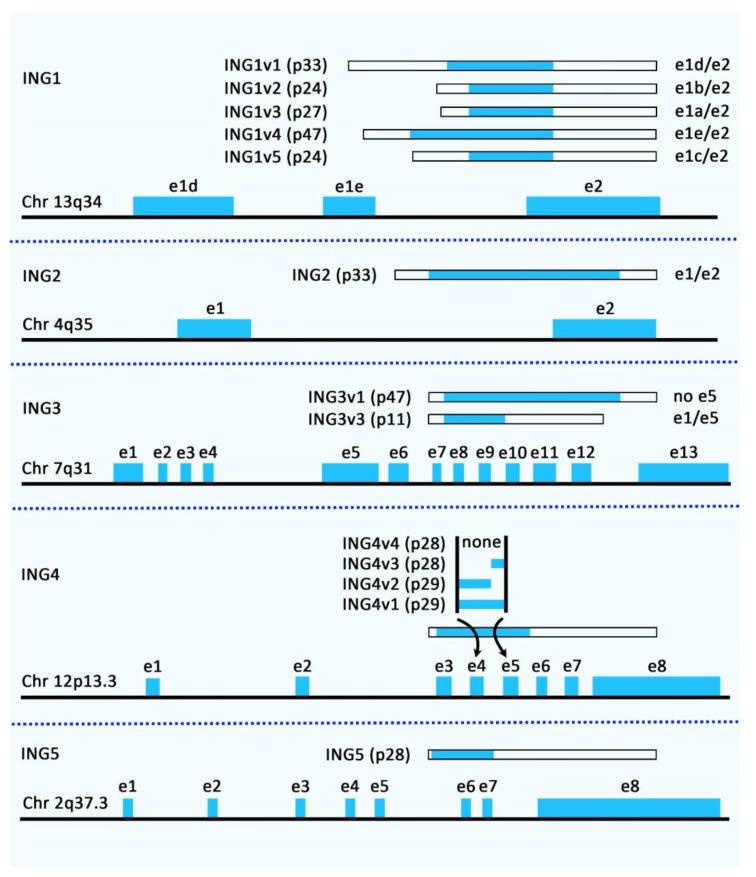
Genomic organization and alternatively spliced variants of human ING family members.

**Figure 2 cancers-14-03107-f002:**
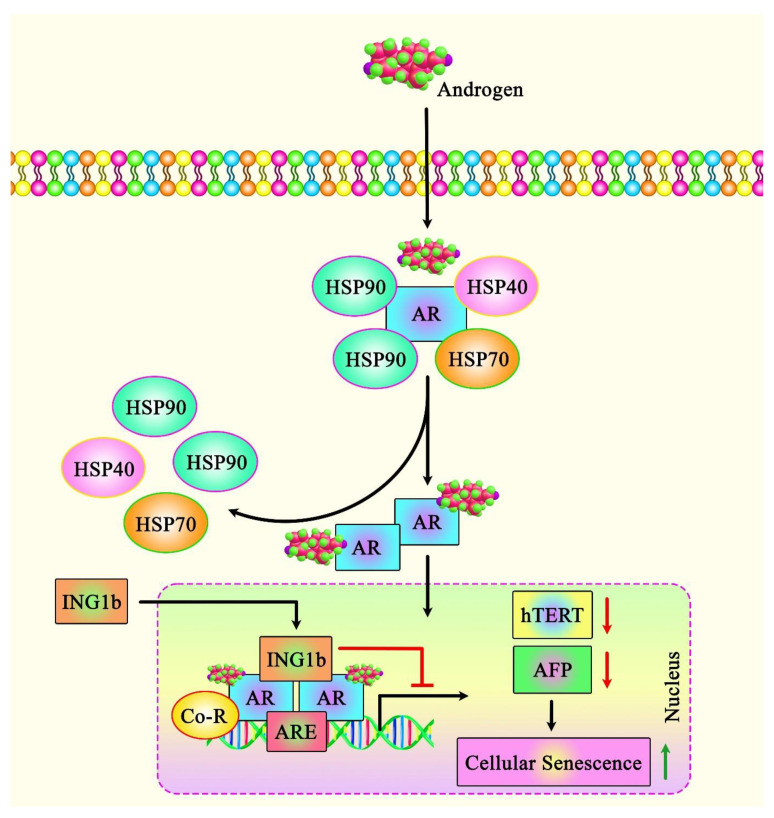
A schematic illustration of the functional interaction of ING1b with AR in prostate cancer (PCa). A recent study revealed that the tumor suppressor ING1b could play an essential role in attenuating cellular senescence in PCa cells by modulating AR signaling. In fact, ING1b could inhibit AR-responsive promoters as well as endogenous key AR target genes, including hTERT and AFP, thereby reducing tumor cell growth and migration [27].

**Figure 3 cancers-14-03107-f003:**
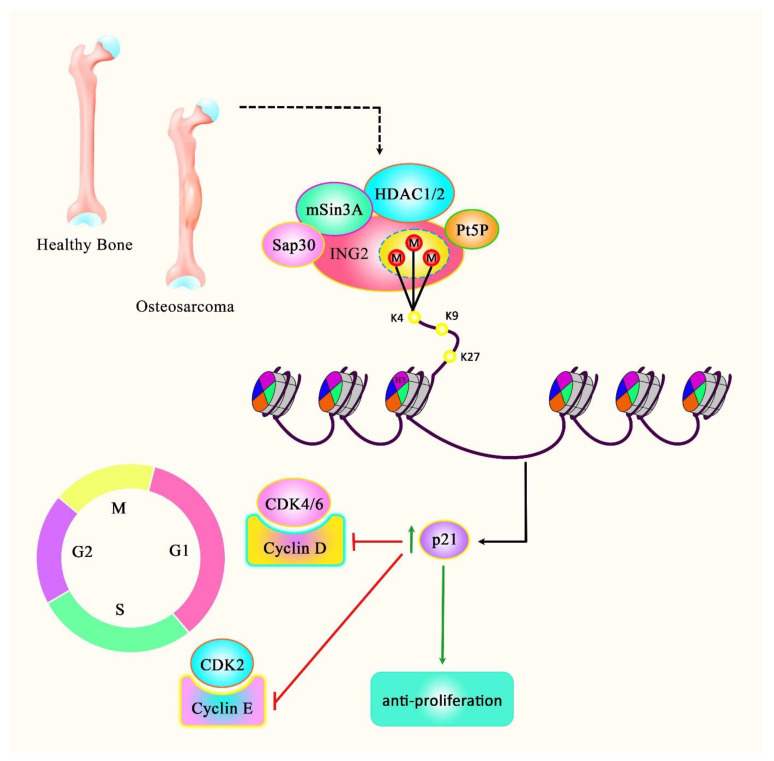
A schematic diagram of the ING2 protein involved in osteosarcoma. Accumulating evidence has detected that ING2 could contribute to enhancing apoptosis, G1 phase arrest, and senescence. Current research has demonstrated that ING2 acts as a tumor suppressor in osteosarcoma via modulating cell cycle progression and cell proliferation [32].

**Table 1 cancers-14-03107-t001:** Summary of information regarding ING family members.

ING Member	Specific Function	Redundant Function	Related Pathways	Reference
ING1	Down-regulation of cyclin B1 and possibly cyclin E, prevention of cell transformation, and induction of apoptosis in cooperation with c-Myc	Regulation of G1/S and G2/M cell cycle transition, regulation of the p53 pathway, chromatin remodeling, and regulation of nucleotide excision repair (NER) in response to UV exposure	NF-κB, ARFMDM2-p53	[1,2]
ING2	Regulation of senescence, regulation of Fas expression, and regulation of apoptosis in response to UV exposure	Regulation of the p53 pathway, regulation of NER, chromatin remodeling, and regulation of the cell cycle and apoptosis	NF-κB, TGFβ	
ING3	-	Regulation of the cell cycle and apoptosis, chromatin remodeling and neural development	-	
ING4	Inhibition of colony formation, reduction of the percentage of cells in S phase, up-regulation of Bax expression, angiogenesis, and cell migration	Regulation of the p53 pathway and chromatin remodeling	NF-κB, HIF-1α	
ING5	Reduction of colony-forming efficiency, inhibition of S-phase, and induction of p21	Regulation of the p53 pathway, chromatin remodeling, and regulation of the cell cycle and apoptosis	-	

**Table 2 cancers-14-03107-t002:** Impact of INGs on cellular senescence based on cell line studies.

Disease/Cellular Mechanism	ING Type and Signaling Pathways	Cell Line	Function	Reference
Aging	ING1a and Rb pathway	Hs68 fibroblast cells, and EA.hy926 and HaCaT cells	↑↑ ING1a: ↑ senescence in the absence of activating p53-mediated DNA damage signaling by activating the Rb pathway	[20]
Breast cancer	ING5	MDA-MB-231 and MCF-7 cells	↑↑ ING5: ↓ cell viability, glycolysis, and mitochondrial respiration, ↑ apoptosis, S arrest, autophagy, or senescence, and ↑ chemoresistance to Cisplatin, MG132, paclitaxel, and SAHA	[44]
Glioma	ING5	U87	↑↑ ING5: ↓ proliferation, energy metabolism, migration, and invasion, and ↑ G2/M arrest, apoptosis, dedifferentiation, and senescence	[45]
Head and neck squamous cell carcinoma	ING3, p300, p21, and p53	HNSCC cells	↑↑ ING3: ↑ p53-mediated cell-cycle arrest, senescence, and/or apoptosis via interacting with p300 and p21	[38]
Induction of apoptosis	ING1 and BAX	SKBR3; MDA-MB468, BT474, and T47D lines; HCT116, and HEK293	The degree of mitochondrial translocation is correlated with the ability of ING1 to induce apoptosis. Binding and activation of BAX by ING1 are needed for the induction of apoptosis.	[25]
Osteosarcoma	ING2	HOS cells	↑↑ ING2: ↑ apoptosis, G1 phase arrest, and senescence	[32]
Ovarian cancer	ING5	ES-2, H08910, H08910-PM, OVCAR3, SKOV3/DDP, A2780, and A2780/T	↑↑ ING5: ↓ cell viability and migration, invasion, and ↑ apoptosis, cell cycle arrest, senescence, and autophagy	[46]
Prostate cancer	ING1b, ING2, and AR signaling	LNCaP PCa cells	∆ ING1b: ↓ growth and migration, ↑ induction of cellular senescence and the cell cycle inhibitor p16 INK4a ↑↑ ING2: causes ↑ growth arrest, and induces cellular senescence by interacting with AR and inhibiting AR transcriptional activation	[26]
	ING1b and AR signaling	ING1b knockout (KO) mouse embryonic fibroblasts (MEFs), PC3 cells, PC3-AR cell line, C4-2 cells, LNCaP, and NIH3T3 S2-6	↑↑ ING1b: ↑ cellular senescence, and ↓ growth and migration via inhibiting AR-mediated transactivation	[27]
Regulation of endocytosis	ING1a, Rb-E2F pathway, and ITSN2	Hs68 and WI38 fibroblast cells	↑↑ ING1a: ↑ cellular senescence to regulate endocytosis via the Rb-E2F pathway and overexpression of ITSN2	[28]
Tongue squamous cell carcinoma	ING5, two truncated fragments of ING5 (aa 1-184 and aa 107-226), cyclin E, and CDK2	HSC-3	↑↑ ING5: ↓ proliferation and ↑ apoptosis in HSC-3 cells Two truncated fragments of ING5 (aa 1-184 and aa 107-226): ↑ cellular senescence and inhibition of cyclin E and CDK2 expression.	[47]

↑ up regulation, ↓ down regulation. ∆ knockdown.

**Table 3 cancers-14-03107-t003:** Role of INGs in cellular senescence based on animal studies (∆: knock-down, deletion).

Tumor Type	Animal Models	Results	Reference
Breast cancer	Balb/c nude mice	↑↑ ING5: ↓ tumor volume and weight	[44]
Glioma	Female Balb/c nude mice	↑↑ ING5: ↓ tumor volume and weight	[45]
Ovarian cancer	Female Balb/c nude mice	↑↑ ING5: ↓ proliferation, and ↑ apoptosis and autophagy	[46]
Prostate cancer	Ing1 knockout (KO) mice	∆ ING1: ↓ endogenous AR target genes	[26]

↑ up regulation, ↓ down regulation.

## Data Availability

Not applicable.

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
