# Peer review of "Inhibitor of Growth Factors Regulate Cellular Senescence"

_cancers, 2022, doi:10.3390/cancers14133107_

Round 1

Reviewer 1 Report

In the review entitled “Inhibitor of Growth Factors Regulate Cellular Senescence”, the authors summarise the current evidences on the role of the Inhibitor of Growth (ING) protein family in cellular senescence based on the evidence obtained from cell line and animal studies, particularly in the context of cancer.

The review is clear and simple to read and well-organized in paragraphs However, it can’t be considered for publicaton in its present form but it needs to be deeply revised in some points, as I reported here:

First of all, the authors should introduce ING family members through a schematic representation of both transcripts and proteins evidencing exons involved in splicing events and conserved domains of the different protein members. Then, it would be important to show the chromosomal localizations of genes and correlate them with potential structural changes that lead to tumor onset and progression.

The review lacks of many references throughout the manuscript that have to be added within the text, in order to support and complete authors’ comments to INGs functions and related pathways.

The authors should also add a table showing the different ING family members, their specific and redundant functions, the molecular pathways in which they are involved and the correlation between cancers and their altered expression.

The authors should also add a summary with comments to ING family members in general, that would be useful to introduce to paragraph 2.

The authors should also add schematic representations of functional activity for each members of ING family (at the moment they are reported only for ING1 ad ING2).

The authors should extend “Animal studies” comments to support data reported and show the In vivo role of ING family members.

Finally, the authors have to correct typing errors within the text and to revise English grammar.

In conclusion, I think that the review can be considered for publication after the revision suggested above and a litte English grammatycal restyling.

Author Response

We thank the reviewers for their fruitful comments and have addressed these as follows:

Reviewer 1

Thank you for rating our review is clear and simple to read and well-organized in paragraphs.

First of all, the authors should introduce ING family members through a schematic representation of both transcripts and proteins evidencing exons involved in splicing events and conserved domains of the different protein members. Then, it would be important to show the chromosomal localizations of genes and correlate them with potential structural changes that lead to tumor onset and progression.

Response: We added a Figure showing the genomic organization and alternatively spliced variants of human ING family members.

The review lacks of many references throughout the manuscript that have to be added within the text, in order to support and complete authors' comments to INGs functions and related pathways.

Response: We added more references.

The authors should also add a table showing the different ING family members, their specific and redundant functions, the molecular pathways in which they are involved and the correlation between cancers and their altered expression.

Response: Although this review is focused on cellular senescence, we added a Table showing different ING family members, their specific and redundant functions and related molecular pathways.

The authors should also add a summary with comments to ING family members in general, that would be useful to introduce to paragraph 2.

Response: We added a paragraph that summarizes information about ING family members.

The authors should also add schematic representations of functional activity for each members of ING family (at the moment they are reported only for ING1 and ING2).

Response: Since the molecular mechanism of participation of other members of ING family in cellular processes is not so clear, we could not make such figures for them.

The authors should extend "Animal studies" comments to support data reported and show the In vivo role of ING family members.

Response: We extended "Animal studies" part.

Finally, the authors have to correct typing errors within the text and to revise English grammar.

Response: We edited the language of the manuscript.

Reviewer 2 Report

The review generally reads well, but lacks in depth and misses several key references.

- Line 13, ING3 is not found in mSIN3a complexes (see doi: 10.1128/MCB.23.10.3456-3467.2003).

- Similarly, line 26-27, should reference the work of Doyon et al (doi.org/10.1016/j.molcel.2005.12.007). Generally accepted that ING1 and ING2 are mSIN3a/HDAC complex subunits, while ING3-5 are subunits of HAT complexes.

- Should spell out UV on line 70.

- Line 81, which ING protein interacts with mitochondrial proteins?

- Line 110-111, "Expression of Intersectin 2 has IS"

- I would advise using a bit less colours in figures. The rainbow colouring are overused.

- Line 142 the work of PMID: 16728974 and 16728977 should be referenced.

- Line 169, spell out PCa (prostate cancer). Also, the work of McClurg et al doi: 10.1038/bjc.2017.447 should be referenced.

- Lines 206-207, What does it mean? Upregulation of ING5 also induces senescence. Is there a domain that is required for induction of senescence?

- Section 2.6 (Animal studies) could be extended easily. Knockouts for Ing2, Ing3, and Ing4 are not mentioned.

Author Response

We thank the reviewers for their fruitful comments and have addressed these as follows:

Reviewer 2

The review generally reads well, but lacks in depth and misses several key references.

- Line 13, ING3 is not found in mSIN3a complexes (see doi: 10.1128/MCB.23.10.3456-3467.2003).

Response: We corrected this point.

- Similarly, line 26-27, should reference the work of Doyon et al (doi.org/10.1016/j.molcel.2005.12.007). Generally accepted that ING1 and ING2 are mSIN3a/HDAC complex subunits, while ING3-5 are subunits of HAT complexes.

Response: We corrected the mentioned point about ING3 and referred to Doyon et al. study in this paragraph (reference 9).

- Should spell out UV on line 70.

Response: We spelled out UV.

- Line 81, which ING protein interacts with mitochondrial proteins?

Response: ING1. We corrected this note.

- Line 110-111, "Expression of Intersectin 2 has IS"

Response: We corrected this grammatical point.

- I would advise using a bit less colours in figures. The rainbow colouring are overused.

Response: We edited the coloring of figures.

- Line 142 the work of PMID: 16728974 and 16728977 should be referenced.

Response: We added these references.

- Line 169, spell out PCa (prostate cancer). Also, the work of McClurg et al doi: 10.1038/bjc.2017.447 should be referenced.

Response: We have spelled out PCa in line 106. We referred to the mentioned reference.

- Lines 206-207, What does it mean? Upregulation of ING5 also induces senescence. Is there a domain that is required for induction of senescence?

Response: We modified this section and omitted the mentioned sentence. There was no data regarding the presence of a domain that is required for induction of senescence in the original article.

- Section 2.6 (Animal studies) could be extended easily. Knockouts for Ing2, Ing3, and Ing4 are not mentioned.

Response: We added more points to this section.

Round 2

Reviewer 1 Report

In the 2nd submission of the review entitled Inhibitor of Growth Factors Regulate Cellular Senescence”, the authors exhaustively responded to my comments.

With respect to my comments, I think they completely responded and added interesting and clarifying images to the manuscript, and the paper results clearer than in the first version.

The only concern regards Figure 1. Genomic organization and alternatively spliced variants of human ING 32 family members. I’m not able to see it. Please verify it and eventually substitute the wrong file with the correct one.

The paragraph are clear and functional to the review.

I think the review meets the journal aims and can be considered for publication (after Figure 1 substitution).

Author Response

In the 2nd submission of the review entitled Inhibitor of Growth Factors Regulate Cellular Senescence”, the authors exhaustively responded to my comments.

Response: Thank you very much.

With respect to my comments, I think they completely responded and added interesting and clarifying images to the manuscript, and the paper results clearer than in the first version.

The only concern regards Figure 1. Genomic organization and alternatively spliced variants of human ING 32 family members. I’m not able to see it. Please verify it and eventually substitute the wrong file with the correct one.

The paragraph are clear and functional to the review.

I think the review meets the journal aims and can be considered for publication (after Figure 1 substitution).

Response: Thank you for your comment. Now, Figure 1 is placed in the correct position in the manuscript.

Reviewer 2 Report

line 36, ING3 associates with TIP60 complex, a histone acetyltransferase. Note that although ING1 and ING2 can associate with HAT activity [doi: 10.1016/j.yexcr.2007.02.010 and doi: 10.1128/MCB.25.15.6639-6648.2005], they are mainly found in mSIN3A/HDAC complex [Doyon et al].

line 171, of PHD

Author Response

line 36, ING3 associates with TIP60 complex, a histone acetyltransferase. Note that although ING1 and ING2 can associate with HAT activity [doi: 10.1016/j.yexcr.2007.02.010 and doi: 10.1128/MCB.25.15.6639-6648.2005], they are mainly found in mSIN3A/HDAC complex [Doyon et al].

Response: We added this point. See page 2.

line 171, of PHD

Response: We corrected this point.